# Silk Sericin: A Promising Sustainable Biomaterial for Biomedical and Pharmaceutical Applications

**DOI:** 10.3390/polym14224931

**Published:** 2022-11-15

**Authors:** Andreia S. Silva, Elisabete C. Costa, Sara Reis, Carina Spencer, Ricardo C. Calhelha, Sónia P. Miguel, Maximiano P. Ribeiro, Lillian Barros, Josiana A. Vaz, Paula Coutinho

**Affiliations:** 1Centro de Investigação da Montanha (CIMO), Instituto Politécnico de Bragança, Campus de Santa Apolónia, 5300-253 Bragança, Portugal; 2Laboratório Associado para a Sustentabilidade e Tecnologia em Regiões de Montanha (SusTEC), Instituto Politécnico de Bragança, Campus de Santa Apolónia, 5300-253 Bragança, Portugal; 3CPIRN-IPG—Center of Potential and Innovation of Natural Resources, Polytechnic Institute of Guarda, 6300-559 Guarda, Portugal; 4CICS-UBI—Health Sciences Research Centre, University of Beira Interior, 6200-506 Covilhã, Portugal

**Keywords:** *Bombyx mori*, silk sericin, biomaterials, biomedical and pharmaceutical applications, tissue engineering, drug delivery systems

## Abstract

Silk is a natural composite fiber composed mainly of hydrophobic fibroin and hydrophilic sericin, produced by the silkworm *Bombyx mori.* In the textile industry, the cocoons of *B. mori* are processed into silk fabric, where the sericin is substantially removed and usually discarded in wastewater. This wastewater pollutes the environment and water sources. However, sericin has been recognized as a potential biomaterial due to its biocompatibility, immunocompatibility, biodegradability, anti-inflammatory, antibacterial, antioxidant and photoprotective properties. Moreover, sericin can produce hydrogels, films, sponges, foams, dressings, particles, fibers, etc., for various biomedical and pharmaceutical applications (e.g., tissue engineering, wound healing, drug delivery, cosmetics). Given the severe environmental pollution caused by the disposal of sericin and its beneficial properties, there has been growing interest in upcycling this biomaterial, which could have a strong and positive economic, social and environmental impact.

## 1. Introduction

Silk is a natural composite fiber produced by the silkworm *Bombyx mori* (*B. mori*) to assemble the cocoon that provides the ideal conditions for the larvae to metamorphose into adults. In 1865, the chemist Emil Cramer reported that the two main components of silk are the hydrophobic fibroin and the hydrophilic sericin [1]. The sericin binds and coats the two fibroin filaments in the raw silk. Although it is known that the textile industry has been using silk for more than 5000 years [2], there is evidence that humans were using silk as early as 8500 years ago [3]. In the textile industry, sericin is removed from the raw silk (degumming process), rendering a much finer silk fiber with a better luster and texture, which is used to make yarns and fabrics [4]. Global silk production in 2018–2019 was 192,692 metric tons [5], with China and India being the main producers [6]. Since sericin is the second largest component of raw silk (after fibroin), it is estimated that out of 400,000 tons of dry cocoons produced worldwide, 50,000 tons of sericin are (usually) discarded in the effluent [7], causing environmental problems [8,9]. The deposited organic load in aqueous effluents with chemical oxygen demand can deplete oxygen in water systems, leading to eutrophication and thus threatening aquatic life [8,9,10].

Sericin, removed during the degumming of silk, is considered waste or by-product despite its biological properties such as biocompatibility, immunocompatibility, biodegradability, anti-inflammatory, antibacterial, antioxidant and photoprotective, among others (as previously reviewed [7,11,12,13,14,15,16,17] and represented in Figure 1). This protein has been recognized as a potential sustainable biomaterial for various biomedical and pharmaceutical applications [7,11,12,13,14,15,16,17,18,19,20,21,22,23].

Indeed, sericin is a highly hydrophilic molecule consisting of polar hydroxyl, carboxyl, and several amino groups [12]. The organic composition, solubility and structural organization of these polar chemical groups are responsible for the biological properties but also allow the formation of blends with other polymers through crosslinking, copolymerization or blending, improving the mechanical resistance of sericin-based biomaterials [24]. When sericin is crosslinked or combined with other polymers, it can be incorporated into hydrogels, films, sponges, particles and fibers with particular properties relevant to biomedical applications [25,26,27], such as tissue engineering [14,28], wound dressings [29] and drug delivery systems [28,30], and to pharmaceutical applications [31].

Given the severe environmental pollution caused by the disposal of sericin and because of its beneficial properties, interest in the recovery of sericin has increased. Thus, the valorization of sericin for biomedical and pharmaceutical applications could have a substantial economic, social, and environmental impact. This article reviews the properties of sericin and recent advances in the use of this protein for biomedical and pharmaceutical applications, focusing on the development of smart drug delivery systems and reported clinical trials using sericin-based products for tissue engineering.

## 2. Biosynthesis and Genetics of Sericin and Its Physical and Chemical Properties

Along with cotton, wool, linen and hemp, silk is one of the most abundant naturally derived fiber. Silk is produced by various animals, such as spiders (*Nephila clavipes* and *Araneus diadematus*), domestic silkworms (*B. mori*) and wild silkworms (*Antheraea pernyi* and *Samia cynthia ricini*) [32]—Figure 1. As the availability of spiders is limited, mainly domestic silkworms are used to obtain silk for the production of textile products—sericulture. For this purpose, silkworm eggs of *B. mori*, a holometabolous insect belonging to the Lepidoptera order and *Bombycidae* family, are laid and incubated before the larvae hatch [32]. Afterwards, the larvae are fed with mulberry leaves for six weeks. Then, until the end of the fifth larval instar stage, the larvae form the silk cocoons that protect them during metamorphosis and create the necessary conditions for larval metamorphosis into adults [4,12]. Before the silkworms turn into pupae, they are sacrificed, and the cocoons are recovered for silk extraction. The killing of the silkworm is necessary to preserve the quality and length of the fiber, as it digests the cocoon as a way out. The finished *B. mori* silk consists of various types of chemical components, mostly fibroin (70–80%) and sericin (20–30%), as well as others (carbohydrates (1.2–1.6%), inorganic matter (0.6–0.7%), wax matter (0.4–0.8%) and pigments (0.2–0.3%)) [15,33]. Figure 2 illustrates the chemical structure of silk polymer and the intermolecular hydrogen bonds between fibroin and sericin [34]. Fibroin acts as the inner core and gives the fiber mechanical strength, while sericin is the outer glue-like coating. Each silk fiber contains two fibroin filaments coated with sericin. Both fibroin and sericin are constituted by a repeated amino acid sequence capable of forming the β-sheet structure. Fibroin has the sequence [GAGAGS]n, and part of the repeat sequence in the sericin has the sequence GSVSSTGSSSNTDSST, where G, A, S, T, V, N and D denote glycine, alanine, serine, threonine, valine, asparagine and aspartic acid, respectively [34].

Silk sericin is produced in the labial glands of *B. mori*, commonly referred to as silk glands [32,35]. The silk glands are long and paired structures originating from the labial region. They are anatomically and physiologically divided into three major compartments: the anterior silk gland (which forms the excretory duct), the middle silk gland (which secretes three types of sericin) and the posterior silk gland (which secretes fibroin)—Figure 3 [35,36,37].

Sericin is produced by alternative splicing of sericin genes [38,39]. The expression of the genes is temporally regulated depending on larval development, resulting in a certain homogeneity between exons, and it is responsible for the large protein diversity. There are at least three genes responsible for sericin synthesis, *Ser1*, *Ser2*, and *Ser3*. Sericin is a globular protein composed of random coil and β-sheets with a molecular weight ranging from 20 to 400 kDa, which is mainly influenced by the extraction method (described in detail in the next section). Sericin comprises 18 amino acids with 45.8% hydroxy amino acids (serine and threonine), 42.3% polar amino acids and 12.2% non-polar amino acids [40,41,42].

The water solubility of sericin is influenced by its position in the silk fiber, i.e., sericin located in the outer layer of the fiber is most soluble in warm water (α-sericin); while sericin located in the inner layer of the fiber (next to fibroin) is insoluble in hot water (β-sericin) [37].

The gelation process of sericin is possible due to the conversion of the random coil into a β-structure, i.e., the sericin random coil is soluble in hot water. However, at low temperature (10 °C, pH about 6–7), this structure converts to a β-sheet structure which facilitates the formation of a three-dimensional network and promotes the formation of the sericin gel [43,44,45]. This phenomenon is reversible when the sample is heated (50–60 °C) [43,44,45]. On the other hand, gelation of sericin can also be achieved by chemical crosslinking (e.g., with glutaraldehyde), leading to the formation of a stable β-sheet structure.

## 3. Extraction Methods of Sericin from Silk Cocoons

The degumming of silk is the process that leads to the cleavage of the peptide bonds (Figure 2) by the hydrolysis of sericin and the subsequent detachment of sericin from fibroin. The extraction of sericin from silk is possible because fibroin is hydrophobic and insoluble in water, while sericin is hydrophilic and can be dissolved in water. 

The conventional method used by the silk industry to degum silk uses detergents/soaps (e.g., Marseille) and a solution of sodium bicarbonate (Na_2_CO_3_)—Table 1. This method successfully removes sericin from cocoons, allowing the recovery of clean and isolated fibroin that the textile industry can use. However, the recovered sericin is highly degraded, reducing its molecular weight and losing some functional properties [46]. In addition, the separation of soap and sericin is very complex. Consequently, traces of soap may remain in the sericin, limiting its use for biomedical and pharmaceutical purposes [47].

Therefore, other degumming methods have been developed to favor the recovery of sericin instead of fibroin, including heat, chemical and enzymatic methods [28]—Table 1 and Figure 1. All these methods can be adapted in terms of time, temperature, chemical additives and others [12,28,48,49,50]. Sometimes, these methods are combined to obtain sericin with desired yield and properties.

In the method that uses heat, the cocoons are usually heated/boiled in water (in combination with/without high pressure by autoclaving). The high temperature and pressure cause instability of the hydrogen bonds between the hydroxyl groups, allowing water to interact with the hydroxyl groups of polar amino acids and further detachment of sericin and fibroin [51].

Acids (citric, tartaric, succinic, etc.) or bases (sodium carbonate, sodium phosphate, sodium silicate, sodium hydrosulfite, etc.) are used to extract the sericin from silk because these chemicals hydrolyze the sericin by breaking the peptide bonds of the amino acid into small molecules, releasing the sericin into the alkaline or acidic solution, in which sericin is highly soluble [52,53]. For example, sodium carbonate converts the -COOH in the sericin molecules to -COONa^+^, which increases its solubility due to the strong hydration of Na^+^ [54].

Proteolytic enzymes (e.g., alcalase, savinase, degummase, papain, trypsin) have also been used to extract sericin [50]. These enzymes cause the hydrolysis of the peptide bonds of the amino acids between the carboxyl group of lysine/arginine and the amino group (NH_2_) of the neighboring amino acids.

All these methods have advantages and disadvantages, as outlined in Table 1. The use of acids and bases to extract sericin can significantly degrade the protein [52]. Moreover, the sericin extracted using acids and bases must be purified in additional steps to remove the chemical impurities. On the other hand, urea-degradation extraction (with 2-mercaptoethanol) has a lower degradative effect on sericin. With this method, about 95% of the total sericin present in the fiber can be extracted without damage. However, this method is expensive and time-consuming [55], and sericin extracted using urea is highly toxic to cells [56]. Considering these limitations, heat is the most commonly used method for sericin extraction. Although this method also causes some degradation of sericin, especially when high temperatures are used or the solutions are applied during long periods [57], sericin retains its remarkable properties. Moreover, the silk is heated in hot distilled water to which no other chemicals are added so that the sericin is obtained without impurities.

In recent years, new technologies have been developed to extract sericin from silk in a greener, more effective and sustainable manner, such as those that use infrared heat, carbon dioxide supercritical fluid and ultrasounds [28,49]. However, these techniques require the use of extra equipment.

### Influence of the Extraction Method on Sericin Yield and Characteristics

It is important to note that the extraction method of *B. mori* silk sericin affects not only its yield but also its physical, chemical and mechanical properties, which in turn will influence the biological properties of sericin.

The extraction of sericin by urea and by autoclave had higher yields (18.60–23.10% and 17.00–21.27%, respectively) compared to extraction using citric acid and sodium carbonate solutions (8.33–15.19% and 5.93–12.69%, respectively) [58].

Sericin with higher molecular weights (10 to >225 kDa) was obtained when extracted with urea. In contrast, the molecular weights of sericin obtained by acid, alkali and heat degradation were 50–150 kDa, 15–75 kDa, and 25–150 kDa, respectively [56]. Furthermore, extraction with the urea solution was the only method that showed clearer protein bands when analyzed by polyacrylamide gel electrophoresis (SDS-PAGE). This demonstrates that the urea-based extraction method has a lower impact on sericin degradation and thus allows sericin recovery with more defined molecular weights.

The conformational changes in the secondary structure of sericin were also observed—Table 2. Sericin extracted by the conventional method and alkali-degradation showed the presence of α-helix, random coil and turns. In contrast, autoclaving (heat)- extracted sericin lacked α-helical structures [16].

Measurements demonstrated that sericin zeta potential is negative independently of the extraction method used. The zeta potential of sericin from urea extraction yielded the highest negative charge (−68.36 mV), followed by acid-degraded sericin (−32.12 mV), heat-degraded sericin (−20.69 mV) and finally, alkali-degraded sericin (−15.87 mV) [56].

Table 3 summarizes the most significant changes in the amino acid composition of sericin under different extraction methods. As shown by Aramwit et al. [56], regardless of the extraction method, serine was the most abundant amino acid in sericin, followed by aspartic acid and glycine. The amount of methionine found in sericin extracted by heat was significantly higher than in sericin extracted by other methods. In comparison, the amount of tyrosine found in urea-extracted sericin was substantially lower than in sericin extracted by other methods.

Significantly, the content of secondary metabolites (phenols and flavonoids) associated with sericin also differed depending on the extraction methods [16]—Table 4. The total phenol content was higher when sericin was extracted by heat (boiling water) and lower when sericin was extracted with urea solutions. Acid-degraded sericin showed the highest total flavonoid content, while alkali degradation resulted in the lowest flavonoid levels.

It is important to point out that the extraction method, and therefore the properties of sericin, affect the use of this protein. Sericin amino acids assembly (aggregation stands, β-sheets, β-turns formation, etc.) influences cell behavior. In fact, avoiding the chemical degradation of sericin promotes cell growth and attachment due to the arrangement of methionine and cysteine amino acids [59]. The amino acid content also affects sericin’s biological properties (discussed in more detail in the following section) and, thus its performance as a biomaterial. For instance, sericin with higher content of serine and threonine amino acids have higher antioxidant and photoprotective activity. It has also been observed that sericin extracted by different methods affects cell viability and collagen production differently [56]. On the other hand, the molecular weight of sericin affects its application. While low molecular weight (<20 kDa) sericin is generally used in hair care, cosmetics and medications, >20 kDa sericin is preferred for manufacturing products (e.g., drug delivery systems, membranes, hydrogels, fibers) for tissue engineering and other purposes [59].

## 4. Sericin Properties Favorable for Biomedical and Pharmaceutical Applications

The most relevant and promising silk sericin biological properties, such as biodegradability, biocompatibility, immunocompatibility, anti-inflammatory, antibacterial, antioxidant and photoprotective activities, are summarized in Figure 1.

### 4.1. Biocompatibility and Immunological Response

Biocompatibility is the main requirement of any biomedical material. When in contact with the human body, a biomaterial should not induce any adverse effects (e.g., immune response) [60].

Regarding sericin, its biocompatibility has been demonstrated in different works since this protein is an immunologically inert material. The addition of sericin to the culture media of several cell lines has shown that it does not promote cytotoxicity, indicating sericin’s safety to cells [61,62]. Moreover, sericin does not induce immunological responses. In work carried out by Gil et al. [63], it was demonstrated that sericin does not induce macrophage activation. On the other hand, Chlapanidas et al. [64] studied the activity of sericin, obtained from different silkworm strains, in peripheral blood mononuclear cells. The results revealed that sericin (in some strains) promoted a decrease in the in vitro secretion of interferon-gamma (IFNγ). At the same time, no effect was observed on the release of tumor necrosis factor-alpha (TNFα) and interleukin 10 (IL10).

On the other hand, Aramwit et al. [65] evaluated the inflammatory response of a sericin-based cream when applied to induced wounds in rats. The quantification of interleukin 1 beta (IL1β) and TNFα was performed after 7 days of application of the cream. The results demonstrated a significant decrease in the release of these cytokines. This effect was also verified in the works carried out by Dash et al. [18] and Mandal et al. [66], in which sericin decreases the levels of cytokines produced by macrophages and monocytes. Thus, it is possible to confirm that sericin is a biocompatible material which does not provoke severe immune responses.

### 4.2. Biodegradability

A biodegradable biomaterial can be broken down into other substances by the organism through different biological processes. The use of biodegradable biomaterials to produce biomedical products is preferred in most cases since these biomaterials are only maintained in the human body until they serve their purpose, being eliminated gradually and naturally after their degradation. A clear advantage of using biodegradable biomaterials is visible when they are used to treat wounds. Applying a biodegradable wound dressing avoids the need to replace/remove it from the wound site, reducing the pain and discomfort for the patient and the damage to the newly formed tissue [67].

Sericin is a biodegradable polymer, and its degradation is mediated both in vitro and in vivo by proteolytic enzymes (e.g., protease XIV, α-chymotrypsin, proteinase K, papain, matrix metalloproteinases, and collagenase) which act on the amorphous hydrophilic segments of the heavy and light chains of silk [67,68]. The products resulting from the degradation of sericin are amino acids, which are absorbed by the body, not inducing any immune response. In the literature, some works report that materials produced based on sericin fibers are reabsorbed after 6 weeks of implantation in vivo [68].

Concerning in vitro assays, researchers used protease XI as a model enzyme to assess the degradation profile of sericin [67]. However, it is essential to consider that the rate of enzymatic degradation depends on several factors, namely the structural and morphological characteristics of the structures composed of sericin (e.g., fibers, films, sponges), processing conditions, characteristics of the biological environment at the implantation site and the presence of mechanical and chemical stresses [68].

### 4.3. Anti-Inflammatory Activity

Inflammation is one of the phases of the healing process, in which phagocytosis of necrotic tissues and possible contaminants present at the wound site occurs [69]. In addition, at this stage, inflammatory cells secrete cytokines and growth factors that recruit the cells responsible for forming new tissue. However, this phase must be controlled since the uncontrolled, exuberant expression of inflammatory cytokines promotes the expression of metalloproteinases, which are responsible for the degradation of the extracellular matrix. In this sense, biomaterials developed for wound treatment must be able to control the inflammatory process [70].

In general, assays to determine anti-inflammatory activity are often based on the evaluation of the expression/release of inflammatory cytokines (interleukin 1 (IL-1) and tumor necrosis factor-alpha (TNF-α)). As described in the literature, IL-1 and TNF-α are the most important inflammatory mediators. They also induce the expression of adhesion molecules essential for the proliferative phase [70]. Thus, in vitro and in vivo assays have already demonstrated that sericin controls the release of the inflammatory cytokines IL-1β and TNF-α [12].

### 4.4. Antibacterial Activity

A biomaterial has antibacterial properties if it destroys bacteria or suppresses their growth or ability for multiplication [71]. In recent years, infectious disease management has become an increasing challenge for healthcare systems. For instance, the occurrence of infections during the wound-healing process is considered one of the most severe problems in wound care. The presence of microorganisms at the wound site prevents the healing process from normally occurring, causing other more severe complications at the local and systemic level [72]. Therefore, using biomaterials with antibacterial properties is a promising tool for producing biomedical products. 

According to what has been reported in the literature, the antimicrobial activity of sericin may be related to the presence of cysteine in its composition, an uncharged polar amino acid, due to its sulfhydryl groups [73]. In turn, these sulfhydryl groups can form weak hydrogen bonds with oxygen or nitrogen, producing an extremely reactive compound that affects various enzymatic reactions and metabolic functions of microorganisms [74].

The antimicrobial activity of sericin has been demonstrated against Gram-positive and Gram-negative bacteria. Studies carried out by Ahamad and KumarVootla [75] revealed that sericin has antimicrobial effects against *Escherichia coli*, *Staphylococcus aureus* and fungi such as *Candida albicans* and *Aspergillus flavus*. Similarly, Jassim and Al-Saree [76] found that when sericin concentration was increased (10–20 mg/mL), the inhibitory effect on the growth of *Streptococcus pneumoniae*, *Pseudomonas aeruginosa* and *E. coli* was also potentiated. Additionally, the authors also found that the number of colonies was reduced from 146 to 29 after treatment with a sericin solution (2%) for 7 days [76].

In addition to antibacterial activity, silk sericin has been reported as an anti-biofilm agent. Recently, Aramwit and other authors developed an in vitro study to investigate the sericin’s potential to inhibit biofilm formation (prevention) and disrupt already formed biofilm (treatment) [77]. They concluded that sericin extracted by urea, heat or acid degradation was able to inhibit and/or reduce biofilm formation, urea-extracted sericin having the highest antibiofilm activity (against *Streptococcus mutans*). This suggests that sericin can be used as an anti-biofilm agent.

### 4.5. Antioxidant and Photoprotective Activity

Reactive oxygen species (ROS) are formed during normal cellular metabolism but become toxic when present in high concentrations. Free radicals and ROS are unstable and react readily with other groups or substances in the body, leading to cell or tissue injury, and are responsible for many diseases (cancer, cirrhosis, ischemic reperfusion, etc.) [78]. Sericin is well known for its potent antioxidant activity. The antioxidant properties of sericin result from its ROS scavenging activity, as well as inhibition of lipid peroxidation, and anti-tyrosinase and anti-elastase activities, as demonstrated by Kato et al. [79]. Moreover, Li et al. [80] showed that sericin might enhance the activity of antioxidant enzymes such as superoxide dismutase, catalase and glutathione peroxidase. 

The antioxidant activities of sericin are correlated with its high serine and threonine content, whose hydroxyl groups act as chelating trace elements such as copper and iron [26,28]. On the other hand, the pigment molecules (e.g., flavonoids and carotenoids) accumulated in sericin layers may be one of the causes that endow sericin with antioxidant properties and anti-tyrosinase activity. Aramwit et al. demonstrated that sericin obtained from cocoons submitted to the pigment extraction had anti-tyrosinase activity, which was higher than the sericin obtained from cocoons with pigments [58].

Sericin is also reported to have photoprotective activity since it can effectively absorb ultraviolet (UV) radiation and prevent oxidative damage by maintaining redox balance. It was previously reported that the topical delivery of *B. mori* sericin protected female hairless mouse from UVB radiation-induced sunburn and tumor initiation [81]. Sericin possesses various amino-based groups rich in hydrogen, oxygen and nitrogen which facilitate strong absorption of UV light wavelengths under 200 nm [82].

## 5. Sericin Biomedical and Pharmaceutical Applications

Various biomaterials have been used for biomedical and pharmaceutical applications. Generally, naturally derived polymers such as agarose, alginate, cellulose, chitosan, collagen, keratin, sericin, etc. (Table 5) are preferred as they have several advantages over the synthetic polymers (poly (anhydride), poly (caprolactone) (PCL), poly (lactic acid) (PLA), poly (glycolic acid) (PGA), poly (lactic-co-glycolic acid) (PLGA), etc.). The biocompatibility, non-toxicity and biodegradability are the most relevant bioactivities of these polymers, similar to native extracellular matrix (ECM), as extensively reviewed in previous works [83,84,85].

Because of its biological properties, but also considering the environmental problems caused by sericin as a discharged waste of the textile industry, this protein has been studied for different applications in different industries [26,27]—Table 6 and Figure 1. Hereafter, this review will be focused on the major biomedical and pharmaceutical applications of sericin, namely tissue engineering, drug delivery and other applications such as metabolic disorders and cosmetic formulations.

### 5.1. Drug Delivery

Scaffolds, films, hydrogels, fibers, foams, spheres, capsules and microneedles, among others, can be used for local and systemic drug delivery. Since sericin has an amphiphilic character (polar side chains and hydrophobic domains), it can be used as a vehicle because it easily binds charged therapeutic molecules or hydrophobic and hydrophilic drugs [95]. In addition, sericin has a long half-life in vivo and high moisture absorption and desorption abilities, which are also favorable properties for its application for drug delivery purposes [28].

Sericin-based structures, mostly hydrogels, prepared by crosslinking, ethanol precipitation, or blending with other polymers, can be used for drug delivery.

Yan et al. [96] developed a hydrogel using sericin and poly (ethylene glycol) diacrylate (PEGDA) solution in a 1:1 volume ratio. The 20% sericin-containing hydrogel displayed a greater specific surface area and suitable mechanical properties. Additionally, the behavior of the hydrogel in terms of swelling, drug release, and in vitro cytotoxicity demonstrated its appropriateness for drug delivery when loaded with berberine. Further, antibacterial studies against *S. aureus* and *E. coli.* confirmed that the released berberine maintained the antibacterial activity of this natural compound.

In another study, a crosslinked sericin/dextran injectable hydrogel was synthesized to display efficient drug loading and controlled release of both macromolecular (protein enzyme (horseradish peroxidase, HRP)) and small molecular (antitumor drug doxorubicin (DOX)) drugs [97]. Furthermore, the hydrogel could be used as a photoluminescence-trackable drug delivery system since the sericin’s photoluminescence from this hydrogel was directly and stably correlated with its degradation, enabling long-term in vivo imaging and real-time monitoring of the remaining drug.

Further, sericin can be used to produce drug carriers such as nanoparticles and microparticles due to its chemical reactivity that allows the easy binding of molecules [28]. In this context, in 2019, Yalcin et al. [98] developed albumin–sericin nanoparticles modified with poly-L-lysine (PLL) and decorated with hyaluronic acid (HA) as a novel small interfering RNA (siRNA) delivery system for laryngeal cancer treatment. In vitro studies carried out with Hep-2 cells demonstrated that the nanoparticles (with albumin and sericin at a ratio of 2:1 (*w*/*w*)) promoted gene silencing, resulting in significant inhibition of cell growth and inducing the cells’ apoptosis. 

In turn, stable micelles were developed by Deng et al. [99] by conjugating hydrophilic sericin with hydrophobic cholesterol, folic acid (tumor-targeting agent) and a near-infrared dye (IR780 iodide). The results showed that the micelles could be absorbed by folic acid-positive gastric cancer cells (BGC-823) through folic acid receptors. Moreover, the nanoparticles showed remarkable effectiveness in photodynamic and photothermal therapy for cancer cells. 

Recently, in 2022, Xu et al. [100] reported an effective strategy to fabricate a silk sericin nanospheres systems for the delivery of recombinant human lactoferrin (a promising protein to treat ulcerative colitis (UC)). To this end, authors optimized transgenic silkworms to generate genetically engineered silk fibers with significant quantities of recombinant human lactoferrin. The nanoparticle uptake by cells in the inflamed colon of mice was more efficient than that of free lactoferrin in solution. Moreover, a low dose of nanoparticles significantly relieved symptoms of UC in mice and achieved a comparable therapeutic effect to the high dose of free lactoferrin in solution.

Sericin is sensitive to pH due to its strongly polar side groups (hydroxyl, carboxyl, and amino). It has been extensively investigated to produce smart delivery systems, i.e., pH-responsive systems. The use of pH-responsive delivery systems is quite advantageous because they can control the release of the therapeutic compound in an external acidic/alkaline environment, improving its specificity and thus enhancing its efficiency and reducing its side effects [101,102].

Wang et al. [103] produced an injectable hydrogel with sericin (2% *w*/*v*). The hydrogel had excellent cell adhesion capability and effectively promoted cell attachment, proliferation and long-term survival of various cell types (e.g., mouse myoblasts (C2C12) and human keratinocytes (HaCaT)). Compared with alginate hydrogels (2% and 4% *w*/*v*), the sericin hydrogel exhibited higher compressive strength and lower compressive modulus, indicating that the sericin hydrogel has good mechanical properties that allow easy handling during implantation. More importantly, these authors observed that the sericin hydrogel had a pH-responsive character since the maximum degree of swelling was reduced in the acidic environment (pH 3). The swelling behavior of sericin depends on its net charge. Since the isoelectric point of sericin is near pH 4 [104], the net charge of sericin at pH 4 is nearly zero. Consequently, there are equal numbers of negatively and positively charged amino acids. Therefore, the attractive force between negative and positive charges prevents the swelling of sericin molecules. This pH-responsive swelling behavior of the sericin hydrogel makes it useful for delivering therapeutic agents in environments with acidic-neutral pH, such as the microenvironment of tumors. 

In a study performed by Huang et al. [105], folate-conjugated sericin nanoparticles were used for tumor targeting and pH-responsive subcellular delivery of doxorubicin (DOX) to treat cancer. The nanoparticles targeted the folate-receptor-rich human oral epithelium carcinoma cell line (KB). Further, the acid environment of the lysosomes that contained the endocytosed nanoparticles prompted the rapid release of DOX to nuclei.

Hu et al. [106] developed pH-triggered nanoparticles based on silk sericin for enhanced cellular uptake and delivery of DOX. In this work, nanoparticles were prepared by physically reacting the negatively charged sericin with positively charged chitosan. Under mild acidic conditions (e.g., in the tumor microenvironment), the surface charge of the nanoparticles changes from negative to positive charge (due to the increased amino/carboxyl ratio), which improves their cellular uptake due to their high affinity for the negatively charged cell membranes.

Oh et al. [107] also produced a pH-responsive sericin drug delivery system for oral administration. The authors observed that the release of diclofenac (nonsteroidal anti-inflammatory drug) was slightly more efficient at higher pH values (7.4 and 9.2 versus 2.2 and 4). In addition, only 10% of the sericin beads were dissolved at pH 2.2 in the presence of pepsin (proteolytic enzyme in the stomach), whereas 45% of the beads were dissolved at pH 7.4 in the presence of trypsin (proteolytic enzyme in the intestine). Therefore, the beads can be used for oral administration of drugs that cause side effects when released into the stomach.

### 5.2. Tissue Engineering

As already mentioned, sericin-based scaffolds can be used as drug delivery systems and as hydrogels, films, sponges and others for tissue engineering purposes [108,109]. In fact, up to now, sericin is a biomaterial that has been mostly investigated for the regeneration of skin and bone, but also cartilage and adipose tissues, among others [28].

Sericin alone can be used for bone regeneration, which was already evaluated by Noosak et al. [110], where it was verified that sericin could increase the proliferation of osteoblast cells (MC3T3-E1) up to 135%, compared with the untreated control. Nevertheless, for the regeneration of bone tissue, the conjugation of sericin with hydroxyapatite or other calcium phosphate-based materials is the most common blend of biomaterials used to produce scaffolds (reviewed in [111,112]) since these biomaterials enable the mimicking of the organic and inorganic matrixes of the bone, respectively. Furthermore, since sericin has poor mechanical properties, its mixture with other biomaterials is crucial to obtain scaffolds with suitable mechanical properties for bone regeneration. 

Recently, Ming et al. [113] developed a membrane of sericin and hydroxyapatite with osteogenic activity for periodontal bone regeneration. The biocompatible membranes induced the osteogenic differentiation of human periodontal membrane stem cells (hPDLSCs) by activating the expression of osteoblast-related genes (ALP, Runx2, OCN, and OPN) without additional inducers.

In addition to hydroxyapatite, sericin can be conjugated with other biomaterials for bone regeneration. Jiang et al. [114], in 2021, developed an injectable hydrogel for bone regeneration composed of alginate/sericin/graphene oxide. The sericin and graphene oxide contributed synergistically to bone regeneration, i.e., graphene oxide significantly enhanced the spreading, osteogenic differentiation and mineralization of encapsulated rat bone marrow-derived stem/stromal cells (BMSC). In contrast, the sericin promoted the activation of signaling pathways (e.g., MAPK), contributing to the M2 polarization of macrophages that further induced osteogenic differentiation of BMSC cells via several secreted cytokines [114]. Overall, the hydrogel contributed to the successful repair of distal femoral defects in rats.

Silk has been extensively used in suturing incisional wounds and skin injuries since ancient times [115]. Sericin’s inherent property to stimulate cell migration and proliferation seems to be directly related to the accelerated wound healing properties. It is considered a good choice of biomaterial for developing wound dressings and bioartificial skin grafts [116,117,118]. The mitogenic effect of sericin on mammalian cells is well-established in numerous studies, especially on fibroblasts and keratinocytes, which are majorly involved in the wound-healing process [117]. 

Baptista-Silva et al. [119] developed an in situ enzyme-mediated forming sericin hydrogel that demonstrated the potential to regenerate the skin both in vitro and in vivo. The hydrogel was non-toxic to the L929 fibroblast cell line and contributed to cell adhesion, colonization and proliferation. The wounds in diabetic mice treated with the hydrogel for 21 days demonstrated lower granulation tissue and inflammatory cells and a reduction in wound size compared to those treated with a dressing commonly used in the clinic (Tegaderm).

In turn, Sapru et al. [120] produced non-mulberry sericin/chitosan/polyvinyl alcohol (PVA) nanofibrous matrices that supported the adhesion, proliferation and cellular interconnection of human keratinocytes. Furthermore, the membranes could fully regenerate full-thickness wounds. 

Tao et al. [121] demonstrated that the sponges composed of silver nanoparticles–sericin/PVA have the expected high porosity, biocompatibility towards NIH/3T3, HEK-293, and RAW264.7 cells, good wettability, hygroscopicity, mechanical properties and an effective antibacterial activity against *S. aureus*, *E. coli* and *P. aeruginosa*. The following in vivo experiments, using Wistar rats, suggested that the composite dressing could be helpful for re-epithelialization and collagen deposition to promote wound healing, which is crucial for skin regeneration. 

In 2018, Chen et al. [122] prepared a 3D-printed hydrogel scaffold composed of sericin and methacrylic anhydride-modified gelatin (GelMA) by 3D printing and observed that it has the potential to regenerate skin wounds. The authors referred to hydrogel’s transparency as its most notable property, which allowed wound care visualization.

In terms of cardiac tissue regeneration, Song et al. [123] prepared sericin as an injectable hydrogel that, when administrated into an acute myocardial infarction area in mouse models, reduces scar formation and infarct size, increases wall thickness and neovascularization, and inhibits the induced inflammatory responses and apoptosis, thereby leading to a significant functional improvement. Moreover, sericin exhibits angiogenic activity by promoting migration and tubular formation of human umbilical vessel endothelial cells.

To date, the application of sericin for tissue engineering purposes is still mainly investigated in in vitro and in vivo studies. Nevertheless, some clinical studies have been conducted on products that contain sericin. As summarized by Liu et al. [11], some clinical studies have shown the potential of sericin (film, cream, scaffold) for wound healing and skin regeneration [124,125,126,127]. In a clinical study by Aramwit et al. [125], the efficacy of sericin added to a standard antimicrobial cream (silver zinc sulfadiazine) for open wound care was evaluated in the treatment of second-degree burn wounds in a total of 29 patients with 65 burn wounds. The results showed that sericin was safe, as no infection or severe reaction was detected in any wound. Moreover, sericin contributed to wound healing, i.e., the complete healing of the control group (wounds treated with silver zinc sulfadiazine cream only) occurred after 29.28 ± 9.27 days, while it took 22.42 ± 6.33 days in the group treated with sericin. 

A clinical trial performed by Siritientong et al. [126] also showed that sericin-releasing bioactive wound dressing was more efficient in the treatment of split-thickness skin graft donor sites compared to the clinically available wound dressing Bactigras^®^ (complete healing at 14 ± 5.2 days versus 12 ± 5.0 days), and also reduced significantly the pain. In another clinical study, a sericin-based cream improved hydration and reduced pruritus after 6 weeks in hemodialysis patients with uremic pruritus compared to a cream without sericin [124]. For bone regeneration purposes, to the best of our knowledge, only silk (and not sericin isolated) from the cocoon of the silkworm was investigated in a clinical trial [128]. 

### 5.3. Other Applications

Metabolism corresponds to the various chemical reactions that guarantee the energy and structural needs of the human organism. When there is a change in metabolism, metabolic disease can occur [129]. Sericin is composed of 18 types of amino acids, out of which 8 types of amino acids play a significant role in human metabolic pathways [13].

In 2010, Okazaki et al. [130] analyzed the impact of sericin on sugar and lipid metabolism in mice fed with a high-fat diet. The body weight, nourishment utilization and fat weight were not modified by adding 4% sericin to the eating routine for 5 weeks. Yet, this addition reduced serum concentration of free unsaturated fats, cholesterol, phospholipids, triglycerides, very low-density lipoproteins (VLDL) and low-density lipoprotein (LDL).

A study from 2019 investigating the effects of sericin extracted from silkworm *B. mori* cocoon on morphophysiological parameters in mice with obesity induced by a high-fat diet suggested that the physiological changes caused by obesity were not 100% reverted by sericin [131]. However, treatment with sericin restored jejunal morphometry and increased lipid excretion in feces in obese mice, suggesting potential anti-obesity effects. 

Another metabolic dysfunction is diabetes mellitus which is a chronic health condition. Several therapies are available to treat diabetes, although they are associated with severe side effects [132]. Thus, the continuing need for more effective and safer antidiabetic agents is evident [133]. 

Several recent studies have attempted to prove sericin’s antidiabetic activity [130]. In 2020, Dong et al. [134] investigated silk sericin’s in vivo hypoglycaemic effect on Type II diabetic mice. The results demonstrated that sericin significantly decreased fasting blood glucose, fasting plasma insulin and glycosylated serum protein levels. The protein also improved oral glucose and insulin tolerance and enhanced antioxidative activities. Sericin could help maintain normal glucose levels, regulate insulin secretion, insulin and lipid metabolism, and inhibit inflammation. Thus, sericin could be developed into a novel functional health food with a significant hypoglycaemic effect. More recently, in 2022, sericin’s effect on liver injury in Type II diabetic rats was investigated [135]. After 4 weeks of dietary supplementation with sericin, the liver masses and organ coefficients of Type II diabetic rats improved compared with those of the rats not fed with sericin. These results demonstrate that sericin might improve glycogen synthesis, accelerate glycolysis, and inhibit gluconeogenesis by enhancing the anti-oxidation capability and reducing inflammatory reactions. Therefore, sericin could potentially be used to develop functional health foods that can lower blood sugar. Furthermore, the ability of sericin to retain water and unfermented fibers may be indicated to improve constipation [12].

Silk sericin has also been described as an ingredient for skin and hair care cosmetic formulations due to its singular physical-chemical composition [136]. The Expert Panel for Cosmetic Ingredient Safety reviewed the safety of hydrolyzed silk and nine other silk protein ingredients, which function primarily as skin and hair conditioning agents and bulking agents in cosmetic products, and concluded that sericin is among the eight silk protein ingredients considered as safe for cosmetics development [137].

The use of sericin in cosmetics, namely in creams and shampoos, increased hydration, elasticity, cleaning with lower irritancy, and anti-aging and anti-wrinkle effects [12]. For example, it was demonstrated that silk sericin has antioxidant and tyrosinase inhibition activity, indicating its whitening potential, which could be a plus to cosmetic applications [138], as well as the protective effects of a hair care treatment based on silk proteins (fibroin, sericin and other proteins) against the damage induced by hair bleaching and coloring [139].

## 6. Conclusions

The cost, availability and resources required to raise silkworms and process silk threaten the future of this industry. Yet, it is often not recognized that various renewable and sustainable by-products can be obtained during silk production, which has considerable economic value. In the textile industry, silk sericin is usually considered a waste which seriously impacts the environment. However, sericin has been recognized as a biomaterial with outstanding potential for biomedical and pharmaceutical applications due to its promising biological activities. Thus, sericin recovery from silk industry wastewaters has great economic, social and environmental importance.

As mentioned in this review, sericin has unique properties that make it a suitable material for several biomedical and pharmaceutical applications. Developing drug delivery systems incorporating sericin in films, nanoparticles, foams, hydrogels, fibers and other materials is one of the most explored strategies to explore and use sericin in biomedicine. Sericin also has been vastly applied in tissue engineering, mainly in bone tissue engineering and wound healing, to promote cell adhesion and proliferation, tissue reconstruction and repair, and skin re-epithelialization. Overall, silk sericin enables the design and development of economically viable, biocompatible and biodegradable products. In addition, the use of sericin is more advantageous compared to the use of other natural biomaterials because it can be easily functionalized (blended with other natural and synthetic polymers) and is approved by the FDA.

Besides all the beneficial effects and possible applications of silk sericin, it presents some limitations that restrict its application for biomedical purposes. Firstly, the selection of the extraction method assures a standard physicochemical profile and improved biological performance, which should be sustainable and scalable to be applied at the industrial level and even consider the potential contribution to the maintenance of the economic competitiveness of the silk industry. Moreover, silk sericin has a high solubility in water and is an animal-derived product, which can restrict its applications in some biomedical fields [140]. Furthermore, silk sericin cannot be used per se as drug delivery systems or scaffolds for tissue engineering due to its rheological and low mechanical properties. Therefore, sericin-based scaffolds with desirable properties are usually prepared in combination with other components [141].

Considering the sustainability challenge and the promising biomedical potential of sericin as well as other wastes from silkworm rearing, sericulture could be turned into biofactories that generate proteins, lipids and polysaccharides in the future for biomedical and pharmaceutical applications. 

## Figures and Tables

**Figure 1 polymers-14-04931-f001:**
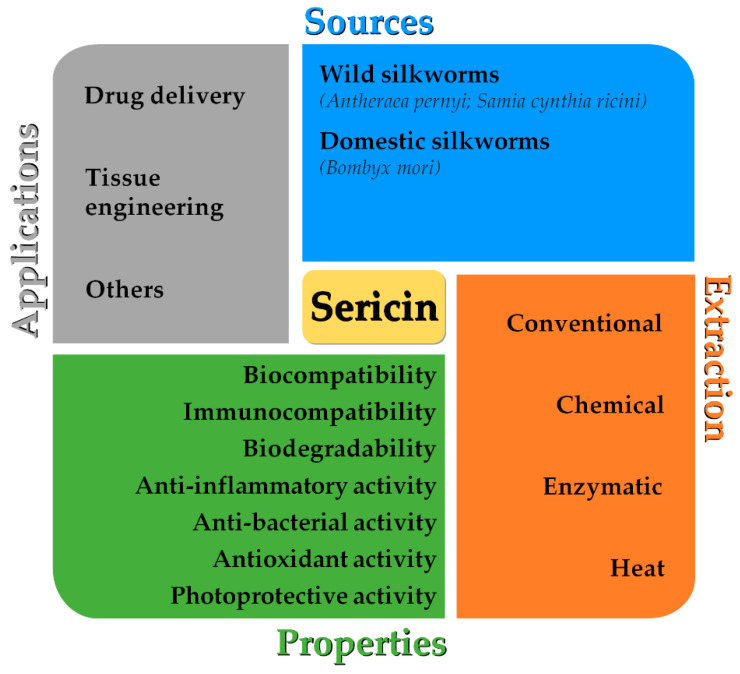
Schematic diagram of sericin sources, extraction methods, properties, and biomedical and pharmaceutical applications.

**Figure 2 polymers-14-04931-f002:**
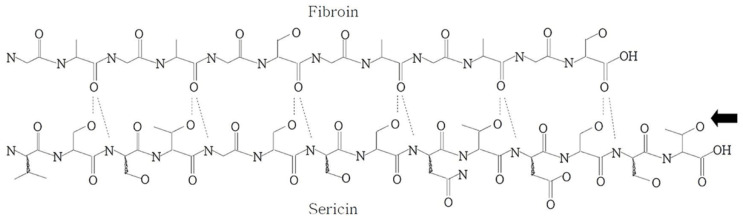
Chemical structure of silk sericin showing the intermolecular hydrogen bonds between the fibroin and sericin. Reprinted with permission from Lee [34]. Copyright © 2022 WILEY-VCH Verlag GmbH & Co. KGaA, Weinheim, Germany.

**Figure 3 polymers-14-04931-f003:**
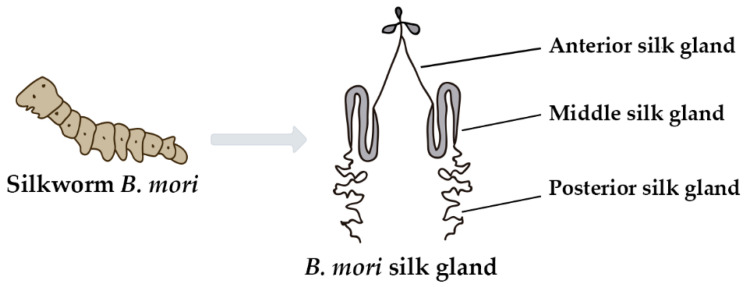
*B. mori* silk gland: anterior silk gland, middle silk gland and posterior silk gland.

**Table 1 polymers-14-04931-t001:** Overview of the main methods used to extract sericin from *B. mori* cocoons.

Extraction Method	Approach	Advantages	Limitations	Ref.(s)
Conventional	Detergents/soaps (e.g., Marseille) and sodium bicarbonate	Effective	Sericin is highly degraded Sericin recovery is difficult It is not environment-friendly/effluent problems	[49]
Chemical	Alkaline solutions (e.g., sodium carbonate, sodium phosphate, sodium silicate, and sodium hydrosulfite)	Quick Low-cost Efficient	Sericin is degraded Sericin recovery is difficult Purification steps are needed It is not environment-friendly/effluent problems	[28,49]
	Acidic solutions (e.g., citric, tartaric, succinic acid).	Sericin is less degraded than when using alkaline solutions	Sericin is degraded Not efficient Purification steps are needed It is not environment-friendly/effluent problems	[28,49]
	Urea (with or without mercaptoethanol)	Effective Time-consuming Sericin is poorly degraded	Purification steps are needed to remove the chemical impurities Toxic to cells	[28,49]
Enzymatic	Proteolytic enzymes (e.g., bromelain, pancreatin, alcalase, savinase, degummase, papain, trypsin, etc.)	Effective Environment-friendly/no effluent problems	Expensive Sericin is degraded Time-consuming	[28,49,50]
Heat	Boiled in water (associated or not with high pressure by autoclaving)	Simple Low-cost Time-consuming No purification steps needed Environment-friendly/no effluent problems	Sericin is degraded (when used at high temperatures) Damages fibroin Removes only the outer layer of sericin	[28,49]

**Table 2 polymers-14-04931-t002:** Percentage (%) of secondary structures (α-helix, β-sheet, turns and random coil) of silk sericin from *B. mori* extracted through different methods. Adapted from [16].

Extraction Method	Secondary Structure (%)			
	α-Helix	β-Sheet	Turns	Random Coils
Conventional	28.8	0.0	35.1	36.1
Heat (boiling in water)	0.0	56.2	2.5	41.3
Urea-degradation	2.8	54.5	4.0	38.7
Alkali-degradation	28.5	0.0	33.8	37.8
Acid-degradation	14.9	34.8	17.0	33.3

**Table 3 polymers-14-04931-t003:** Composition of the amino acid (in mole%) of silk sericin from *B. mori* extracted using various methods. Adapted from [56].

Amino Acid	Extraction Method			
	Heat	Urea-Degradation	Acid-Degradation	Alkali-Degradation
**Serine**	33.63	31.27	31.86	30.01
**Aspartic acid**	15.64	18.31	15.93	19.88
**Glutamic acid**	4.61	5.27	5.75	5.93
**Glycine**	15.03	11.23	10.49	11.01
**Histidine**	1.06	3.26	2.47	1.72
**Arginine**	2.87	5.41	4.92	4.92
**Threonine**	8.16	8.36	8.51	6.49
**Valine**	2.88	2.96	2.95	2.94
**Methionine**	3.39	0.12	0.06	0.15
**Lysine**	2.35	3.14	3.48	2.89
**Isoleucine**	0.56	0.96	0.87	0.75
**Leucine**	1.00	1.58	1.43	1.56
**Phenylalanine**	0.28	0.60	0.71	0.81

**Table 4 polymers-14-04931-t004:** Total phenol and flavonoid content of silk sericin from *B. mori* extracted through different methods. Adapted from [16].

Extraction Method	Total Phenol Content (mg GAE/10 g)	Total Flavonoid Content (mg CE/10 g)
Conventional	253.40 ± 9.10	328.79 ± 47.81
Heat (boiling in water)	319.40 ± 5.70	381.00 ± 47.45
Urea-degradation	200.23 ± 13.50	539.93 ± 46.8
Alkali-degradation	257.73 ± 12.00	210.01 ± 30.09
Acid-degradation	256.07 ± 12.37	708.80 ± 54.49

**Table 5 polymers-14-04931-t005:** Naturally occurring polymers: origin and major advantages/disadvantages for biomedical and pharmaceutical applications.

Polymer	Origin	Advantages	Disadvantages	Ref.(s)
Agarose	Purified from agar that is obtained from red seaweed (e.g., *Ahnfeltia plicata*, *Gelidium amansii*, *Eucheuma*)	Biocompatible Non-immunogenic Thermo-reversible behavior Easy gelling Inexpensive	Soluble only at high temperatures Low cell adhesion Poorly-degradable in humans	[83,86]
Alginate	Brown seaweed (e.g., *Laminaria*, *Macrocystis*, *Ascophyllum*)	Biocompatible Easy chemical modification Easy gelling Inexpensive	Poorly-degradable in humans Sterilization causes degradation Low cell adhesion Weak mechanical strength.	[83,86,87]
Cellulose	Plants and bacteria	Biocompatible Good mechanical strength Porous stable matrix Inexpensive	Poorly-degradable in humans	[83,88,89]
Chitosan	Deacetylation of chitin that is obtained from crustacean exoskeletons, insects and fungal cell walls	Biocompatible Biodegradable Non-toxic Non-antigenic Non-allergenic Bioactive Inexpensive	Immunogenic Non-soluble at physiological pH Low long-term stability Weak mechanical strength	[87,88,89,90]
Collagen	Connective tissue (e.g., cartilage, bones, tendons, ligaments and skin)	Biocompatible Biodegradable Non-toxic Non-antigenic Non-immunogenic Bioactive Native ECM protein	Viral contamination Low stability Sterilization causes degradation Weak mechanical strength Expensive	[86,88,89]
Fibrin	Converted from fibrinogen that is obtained from blood serum	Biocompatible Biodegradable Non-immunogenic Native ECM protein Bioactive	Viral contamination Rapid degradation Weak mechanical strength	[83,91]
Fibroin	Silk of different animals (e.g., spiders (*Nephila clavipes* and *Araneus diadematus*), silkworms (*B. mori*, *Antheraea pernyi* and *Samia cynthia ricini*))	Biocompatible Biodegradable Bioactive Good mechanical strength Thermostable	Non-soluble in water Expensive	[37,83,91,92,93]
Gelatin	Degraded collagen that is obtained from connective tissue	Biocompatible Biodegradable Non-immunogenic Poorly-antigenic Bioactive Thermal crosslinking Easy gelling	Rapid degradation Weak mechanical strength	[83,87,91]
Hyaluronic acid	ECM of the connective tissue (e.g., cartilage, bones, tendons, ligaments, and skin), synovial fluid and other tissues	Biocompatible Biodegradable Non-immunogenic Native ECM protein Bioactive	Viral contamination Rapid degradation Weak mechanical strength Expensive	[83,86,87]
Keratin	Hair, nails, horn, hoofs, wool and feathers	Biocompatible Biodegradable Bioactive	Weak mechanical strength	[83,93,94]
Sericin	Silk of different animals (e.g., spiders (*Nephila clavipes* and *Araneus diadematus*), silkworms (*B. mori*, *Antheraea pernyi* and *Samia cynthia ricini*))	Biocompatible Biodegradable Bioactive Thermo-reversible behavior/easy gelling	Weak mechanical strength	[12,37,83]
**ECM**—extracellular matrix				

**Table 6 polymers-14-04931-t006:** Silk sericin applications in different industries. Adapted from [26,27].

Industry	Applications
Biomedical and pharmaceutical	Supplement in culture media Antitumor activity Metabolic effects (in the gastrointestinal tract, in the circulatory and immune systems, on lipid metabolism and obesity) Tissue engineering Wound healing Drug delivery Contact lenses Matrix for implants Vehicle for cell amplification Stabilizer in vaccines Skincare: skin elasticity, anti-wrinkle and anti-aging influences, UV protection impact Nailcare: prevents cracks and brittleness, and raises the inherent brightness Haircare: conditioner; prevents hair damage Gel: moisturizing property
Food	To enhance the taste and touch of porridge Prevents browning reactions in a variety of ingredients Antioxidants Mineral absorption is accelerated Additive as a nutrient
Textile	In fabrics to absorb moisture Cleaning fabrics Improved antibacterial activity Fabricated nanofiber UV protection of textiles Medical textiles Nanofibers
Others	Treating industrial wastewater with adsorptive pollutants Air filter products Anti-frosting agent for roads and roofs Artificial leather products Roads and roofs Art pigments

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
