# Peer review of "Silk Sericin: A Promising Sustainable Biomaterial for Biomedical and Pharmaceutical Applications"

_polymers, 2022, doi:10.3390/polym14224931_

Round 1
Reviewer 1 Report
The authors report an overview on silk sercine and some of its applications.
The review is well written and well structured, I think it can be suitable for publication on polymers, eventhough, for sake of completeness, I would suggest to cite some very cerent reviews on the same material, e.g.:
https://doi.org/10.1016/j.biomaterials.2022.121638
- https://doi.org/10.1080/00914037.2020.1785454
https://doi.org/10.1007/s10924-022-02381-w
Author Response
The authors report an overview on silk sercine and some of its applications. The review is well written and well structured, I think it can be suitable for publication on polymers, even though, for sake of completeness, I would suggest to cite some very cerent reviews on the same material, e.g.:
- https://doi.org/10.1016/j.biomaterials.2022.121638
- https://doi.org/10.1080/00914037.2020.1785454
- https://doi.org/10.1007/s10924-022-02381-w
Authors response:
Thank you for your comments. As suggested by the reviewer, the authors cited these recent reviews in complement to others. Please see lines 47 and 49 on page 2 and line 153 on page 4.
Reviewer 2 Report
Reviewer’s Report – For authors
ID: polymers-1973912
There are problems with this submission as discussed below.
#1. This manuscript has been submitted for possible publication in the special issue titled “Intranasal Delivery of Novel Polymeric Pharmaceutic Formulations and Implants”. However, there is no mention in the manuscript that silk sericin has ever been proposed or used as a component in nasal formulations or implants, and there is no suggestion made by the authors in the conclusions that such application could be considered. Therefore, there is absolutely no justification for this submission to be included in this special issue.
#2. There is extensive review literature on sericin and its biomedical applications. I am inserting below a list of no less than 11 major reviews published since 2008 onwards (and of course could be more).
[1] Kundu, S.C.; Dash, B.C.; Dash, R.; Kaplan, D.L. Natural protective glue protein, sericin bioengineered by silkworms: potential for biomedical and biotechnological applications. Prog. Polym. Sci. 2008, 33, 998–1012.
[2] Aramwit, P.; Siritientong, T.; Srichana, T. Potential applications of silk sericin, a natural protein from textile industry by-products. Waste Mngt. Res. 2012, 30, 217–224.
[3] Khan, M.M.R.; Tsukada, M. (2014) Electrospun silk sericin nanofibers for biomedical applications. In Silk Biomaterials for TissueEengineering and Regenerative Medicine; Kundu, S.C., Ed.; Elsevier: Amsterdam, The Netherlands, 2014; pp. 125–156.
[4] Cao, T.T.; Zhang, Y.Q. Processing and characterization of silk sericin from Bombyx mori and its application in biomaterials and biomedicines. Mater. Sci. Eng. C 2016, 61, 940–952.
[5] Kunz, R.I.; Costa Brancalhão, R.M.; Chasko Ribeiro, L.dF.; Natali, M.R.M. Silkworm sericin: properties and biomedical applications. BioMed Res. Int. 2016, 8175701.
[6] Rocha, L.K.H.; Favaro, L.I.L.; Rios, A.C.; Silva, E.C.; Silva, W.F.; Stigliani, T.P.; Guilger, M.; Lima, R.; Oliveira Jr, J.M.; Aranha, N.; Tubino, M.; Vila, M.M.D.C.; Balcão, V.M. Sericin from Bombyx mori cocoons. I. Extraction and physicochemical-biological characterization for biopharmaceutical applications. Process Biochem. 2017, 61, 163–177.
[7] Ghosh, S.; Rao, R.S.; Nambiar, K.S.; Haragannavar, V.C.; Augustine, D.; Sowmya, S.V. Sericin, a dietary additive: minireview. J. Med. Radiol. Pathol. Surg. 2019, 6, 4–8.
[8] Suryawanshi, R.; Kanoujia, J.; Parashar, P.; Saraf, S. Sericin: a versatile protein biopolymer with therapeutic significance. Curr. Pharmac. Design 2020, 26, 5414–5429.
[9] Jo, Y.Y.; Kweon, H.Y.; Oh, J.H. Sericin for tissue engineering. Appl. Sci. 2020, 10, 8457.
[10] Arango, M.C.; Montoya, Y.; Peresin, M.S.; Bustamante, J.; Álvarez-López, C. Silk sericin as a biomaterial for tissue engineering: a review. Int. J. Polym. Mater. Polym. Biomater. 2021, 70, 1115–1129.
[11] Elahi, M.; Ali, S.; Tahir, H.M.; Mushtaq, R.; Bhatti, M.F. Sericin and fibroin nanoparticles—natural product for cancer therapy: a comprehensive review. Int. J. Polym. Mater. Polym. Biomater. 2021, 70, 256–269.
Obviously, the authors were not aware of most of these papers, as only two of them were cited (as refs [24] and [97]). This is inadmissible: when you write a review, you must be aware of all previous reviews published on your topic (at least the recent ones). Also, I have difficulties to to figure out what new contributory information could this review possibly bring to the scientific literature about silk sericin in addition to the information contained in the papers listed above. The authors should have cited the previous reviews and explain in the introductory section what informational novelty they are providing with their manuscript. I believe that is none.
#3. The manuscript is written in an English that can be improved both grammatically and syntactically. Inconsistencies in using capitalized words throughout the text, and non-uniform editing of the references list are abundant. Besides, the authors have a habit of using cherry-picked random references to support certain statements, ignoring that those publications were not exactly matching the content of those statements. For instance, the general statement that silk consists of fibroin and sericin (line 31) is supported with refs [1] and [2], recent papers on particular aspects, which definitely were not the first publications to have shown this fact. A major review or book should have been quoted, or – even better – the first paper that reported that finding! The same is valid for their use of ref [2] at line 35 and of ref [3] at line 36. To employ unsuitable references leads to misinformation and it is unethical. The introductory section should be seriously revised.
Author Response
There are problems with this submission as discussed below.
#1. This manuscript has been submitted for possible publication in the special issue titled “Intranasal Delivery of Novel Polymeric Pharmaceutic Formulations and Implants”. However, there is no mention in the manuscript that silk sericin has ever been proposed or used as a component in nasal formulations or implants, and there is no suggestion made by the authors in the conclusions that such application could be considered. Therefore, there is absolutely no justification for this submission to be included in this special issue.
Authors response:
Authors apologize for this incident. There must have been some inopportune problem or miscommunication, as the authors did not intend to submit the manuscript to this specific special issue.
#2. There is extensive review literature on sericin and its biomedical applications. I am inserting below a list of no less than 11 major reviews published since 2008 onwards (and of course could be more).
[1] Kundu, S.C.; Dash, B.C.; Dash, R.; Kaplan, D.L. Natural protective glue protein, sericin bioengineered by silkworms: potential for biomedical and biotechnological applications. Prog. Polym. Sci. 2008, 33, 998–1012.
[2] Aramwit, P.; Siritientong, T.; Srichana, T. Potential applications of silk sericin, a natural protein from textile industry by-products. Waste Mngt. Res. 2012, 30, 217–224.
[3] Khan, M.M.R.; Tsukada, M. (2014) Electrospun silk sericin nanofibers for biomedical applications. In Silk Biomaterials for TissueEengineering and Regenerative Medicine; Kundu, S.C., Ed.; Elsevier: Amsterdam, The Netherlands, 2014; pp. 125–156.
[4] Cao, T.T.; Zhang, Y.Q. Processing and characterization of silk sericin from Bombyx mori and its application in biomaterials and biomedicines. Mater. Sci. Eng. C 2016, 61, 940–952.
[5] Kunz, R.I.; Costa Brancalhão, R.M.; Chasko Ribeiro, L.dF.; Natali, M.R.M. Silkworm sericin: properties and biomedical applications. BioMed Res. Int. 2016, 8175701.
[6] Rocha, L.K.H.; Favaro, L.I.L.; Rios, A.C.; Silva, E.C.; Silva, W.F.; Stigliani, T.P.; Guilger, M.; Lima, R.; Oliveira Jr, J.M.; Aranha, N.; Tubino, M.; Vila, M.M.D.C.; Balcão, V.M. Sericin from Bombyx mori cocoons. I. Extraction and physicochemical-biological characterization for biopharmaceutical applications. Process Biochem. 2017, 61, 163–177.
[7] Ghosh, S.; Rao, R.S.; Nambiar, K.S.; Haragannavar, V.C.; Augustine, D.; Sowmya, S.V. Sericin, a dietary additive: minireview. J. Med. Radiol. Pathol. Surg. 2019, 6, 4–8.
[8] Suryawanshi, R.; Kanoujia, J.; Parashar, P.; Saraf, S. Sericin: a versatile protein biopolymer with therapeutic significance. Curr. Pharmac. Design 2020, 26, 5414–5429.
[9] Jo, Y.Y.; Kweon, H.Y.; Oh, J.H. Sericin for tissue engineering. Appl. Sci. 2020, 10, 8457.
[10] Arango, M.C.; Montoya, Y.; Peresin, M.S.; Bustamante, J.; Álvarez-López, C. Silk sericin as a biomaterial for tissue engineering: a review. Int. J. Polym. Mater. Polym. Biomater. 2021, 70, 1115–1129.
[11] Elahi, M.; Ali, S.; Tahir, H.M.; Mushtaq, R.; Bhatti, M.F. Sericin and fibroin nanoparticles—natural product for cancer therapy: a comprehensive review. Int. J. Polym. Mater. Polym. Biomater. 2021, 70, 256–269.
Obviously, the authors were not aware of most of these papers, as only two of them were cited (as refs [24] and [97]). This is inadmissible: when you write a review, you must be aware of all previous reviews published on your topic (at least the recent ones). Also, I have difficulties to to figure out what new contributory information could this review possibly bring to the scientific literature about silk sericin in addition to the information contained in the papers listed above. The authors should have cited the previous reviews and explain in the introductory section what informational novelty they are providing with their manuscript. I believe that is none.
Authors response:
Authors thank the reviewer for raising this concern about the published review articles on the topic of sericin and its biomedical applications. Authors cited those reviews in the manuscript. Please find the added citations in lines 47 and 49 on page 2.
Please note that the novelty of the current manuscript was explained and clarified at the end of the Introduction section (lines 68 to 71 on page 2). Moreover, section 5. Sericin biomedical and pharmaceutical applications was revised and improved, once is the focus, and the more relevant novelty of this review article. These sub-sections were improved to describe in detail more relevant studies that described sericin's biomedical and pharmaceutical applications. We hope that this modification will contribute not only to the novelty of the manuscript but also to help the readers to learn about the recent progress in this topic. In the “5.1. Drug delivery” sub-section, studies that describe the use of sericin to produce pH-responsive drug delivery systems were added and described since the use of this type of smart systems is an emerging therapeutic approach. On the other hand, in the “5.2. Tissue engineering” sub-section, authors referred to publications abording clinical trials that proved the sericin contribution in the tissue engineering field and thus disclosed the current development stage of sericin-based products. Please see the performed modifications highlighted in sections “5.1. Drug delivery” (lines 390-395 on page 13, lines 434-454 on page 14 and lines 460-474 on page 14) and “5.2. Tissue engineering” (line 542-561 at page 16).
#3. The manuscript is written in an English that can be improved both grammatically and syntactically. Inconsistencies in using capitalized words throughout the text, and non-uniform editing of the references list are abundant. Besides, the authors have a habit of using cherry-picked random references to support certain statements, ignoring that those publications were not exactly matching the content of those statements. For instance, the general statement that silk consists of fibroin and sericin (line 31) is supported with refs [1] and [2], recent papers on particular aspects, which definitely were not the first publications to have shown this fact. A major review or book should have been quoted, or – even better – the first paper that reported that finding! The same is valid for their use of ref [2] at line 35 and of ref [3] at line 36. To employ unsuitable references leads to misinformation and it is unethical. The introductory section should be seriously revised.
Authors response:
The authors acknowledge the reviewer for this observation concerning the unmatching of the content of the statements with the cited articles. Modifications were performed in order to support the statements with adequate references:
The previous references [1] and [2] (previously at line 31) were replaced by:
[1] E. Cramer, Ueber die Bestandtheile der Seide, J für Prakt Chemie 96 (1865) 76-98, DOI: 10.1002/prac.18650960111), which is, to the best of our knowledge, the first publication to describe the composition of the B.mori silk.
The previous references [2] and [3] (previously at lines 35 and 36) were replaced by:
[4] L. Rheinberg, THE ROMANCE OF SILK: A Review of Sericulture and the Silk Industry, Textile Progress 21(4) (1991) 1-43), DOI: 10.1080/00405169108688854
[5] S. Ingle, N. Bagde, R.F. Ansari, A.B. Kayarwar, Analysis of growth and instability of silk production in India, Journal of Pharmacognosy and Phytochemistry 11(4) (2022) 195-201; DOI: 10.22271/phyto.2022.v11.i4c.14464
[6] K.M. Babu, 1 - Silk production and the future of natural silk manufacture, in: R.M. Kozłowski (Ed.), Handbook of Natural Fibres, Woodhead Publishing2012, pp. 3-29, DOI: 10.1533/9780857095510.1.3
Furthermore, as recommended by the reviewer, the citations in the Introduction section and through the manuscript were revised/added/eliminated to only cite publications that adequately support the statements and avoid misinformation. Please see the references highlighted across the manuscript.
Lastly, the text was revised in order to improve the English both grammatically and syntactically, but also to eliminate inconsistencies and non-uniform editing.
Reviewer 3 Report
Comments
In this " Silk sericin: a promising sustainable biomaterial for Biomedical and Pharmaceutical applications", the authors summarise the silk sericin, their extraction methods from silk cocoons and detail the different types of polymeric micelles and their application in biomedical and pharmaceutical applications. The article is generally fine, but still needs some improvement.
My suggestions are as follows.
1. This paper said the “silk is the most abundant natural derived polymers”, how about it chemical structures, its conformity to the characteristics of the polymer needs to be given
2. For a review, I would like to know why there are so few examples of related topics. I feel that some relevant research should be properly depicted, which would help readers to learn more clearly about the content of this review.
3. More examples should be added into the biomedical part, and the comparation with the natural silk peptides with the synthesis polypeptides, comparation with the other kinds of polymers. These references can be cited Biomater. Sci., 2022, 10, 5369-5390 DOI: 10.1039/D2BM00719C. J Control Release. 2022 Aug 6:S0168-3659(22)00485-0. doi: 10.1016/j.jconrel.2022.08.005.
4. Although this article is mainly written about the application of amphiphilic block copolymers in drug delivery, it is too thin in the drug delivery section. Some specific research examples could be added to increase the length of the chapter.
5. In this manuscript, the silk sericin are described and whether these structures will have an impact on drug delivery, tissue engineering, aspects could be expanded.
6. Check the heading. 5.2, 5.4?
Author Response
In this " Silk sericin: a promising sustainable biomaterial for Biomedical and Pharmaceutical applications", the authors summarise the silk sericin, their extraction methods from silk cocoons and detail the different types of polymeric micelles and their application in biomedical and pharmaceutical applications. The article is generally fine, but still needs some improvement.
My suggestions are as follows.
- This paper said the “silk is the most abundant natural derived polymers”, how about it chemical structures, its conformity to the characteristics of the polymer needs to be given
Authors response:
The authors thank the reviewer for this important suggestion. Please note that the chemical structure of the silk polymer was added to the manuscript (new Figure 2 - line 99 on page 3). The chemical composition of the silk and other relevant details were also addressed in the manuscript (please see lines 87-97 on page 3).
- For a review, I would like to know why there are so few examples of related topics. I feel that some relevant research should be properly depicted, which would help readers to learn more clearly about the content of this review.
Authors response:
Thank you for your comments. The authors agree with the reviewer's comment regarding the few examples given for each sub-section of the section “5. Sericin biomedical and pharmaceutical applications”. These sub-sections were improved to describe in detail more relevant studies that described sericin's biomedical and pharmaceutical applications. We hope that this modification will contribute not only to the novelty of the manuscript but also to help the readers to learn about the recent progress in this topic. In the “5.1. Drug delivery” sub-section, studies that describe the use of sericin to produce pH-responsive drug delivery systems were added and described since the use of this type of smart systems is an emerging therapeutic approach. On the other hand, in the “5.2. Tissue engineering” sub-section, authors referred to publications abording clinical trials that proved the sericin contribution in the tissue engineering field and thus disclosed the current development stage of sericin-based products. Please see the performed modifications highlighted in sections “5.1. Drug delivery” (lines 390-395 on page 13, lines 434-454 on page 14 and lines 460-474 on page 14) and “5.2. Tissue engineering” (line 542-561 at page 16).
- More examples should be added into the biomedical part, and the comparation with the natural silk peptides with the synthesis polypeptides, comparation with the other kinds of polymers. These references can be cited Biomater. Sci., 2022,10, 5369-5390 DOI: 10.1039/D2BM00719C. J Control Release. 2022 Aug 6:S0168-3659(22)00485-0. doi: 10.1016/j.jconrel.2022.08.005.
Authors response:
The reviewer raises an interesting point regarding the lack of comparison between silk and silk sericin with other biomaterials. Therefore, in addition to the disclosure of more studies to the biomedical and pharmaceutical section of the manuscript (sub-sections #5.1 to #5.2), as disclosed in the previous author's response, other alterations were performed to comply with the reviewer's comment. Please note that a new table (Table 5) was added to compare sericin with other major natural-derived polymers. Moreover, comparisons of sericin with other biomaterials were added when the described studies performed those studies (please see lines 443-446 on page 14).
Lastly, please see that the authors added the suggested articles to the manuscript (references [103] and [104], line 439 on page 14).
- Although this article is mainly written about the application of amphiphilic block copolymers in drug delivery, it is too thin in the drug delivery section. Some specific research examples could be added to increase the length of the chapter.
Authors response:
In order to increase the length of the “5.1. Drug delivery” sub-section, the authors added and described more studies (as discussed in the author's response to the 2nd reviewer question).
- In this manuscript, the silk sericin are described and whether these structures will have an impact on drug delivery, tissue engineering, aspects could be expanded.
Authors response:
The manuscript was altered to describe better the influence of the sericin structural properties on its application for drug delivery and tissue engineering purposes. So, at the end of the section “3.1. Influence of the extraction method on sericin yield and characteristics”, it was described the influence of the sericin properties (e.g., biochemical properties) in the application of this biomaterial for biomedical and pharmaceutical purposes (please see lines 237-250 at pages 7 and 8). The conclusion section was also altered for the same purpose (please see the highlighted text in the Conclusion section, lines 628-632 on page 17).
- Check the heading. 5.2, 5.4?
Authors response:
The authors thank the reviewer for noticing the typo. The numeration of the sub-sections was rearranged.
Reviewer 4 Report
In the manuscript entitled “Silk sericin: a promising sustainable biomaterial for biomedical and pharmaceutical applications”, the authors have reviewed recent biomaterial research advances of silk sericin, as well provided extensive information about sericin properties. I have only few comments as listed below:
1. In Figure 1, it is mentioned that spiders are one of sericin sources. Please check this because I don’t think spiders produce sericin.
2. Page 12, Line 426, “Silk sercin has been extensively used in suturing…” This looks like a mistake with silk fibroin. Please check it.
3. Page 13, 5.4 should be 5.3.
Author Response
In the manuscript entitled “Silk sericin: a promising sustainable biomaterial for biomedical and pharmaceutical applications”, the authors have reviewed recent biomaterial research advances of silk sericin, as well provided extensive information about sericin properties. I have only few comments as listed below:
- In Figure 1, it is mentioned that spiders are one of sericin sources. Please check this because I don’t think spiders produce sericin.
Authors response:
The authors thank the reviewer for noticing the typo. In fact, spiders like Nephila clavipes and Araneus diadematus produce silk, however their silk does not contain sericin. Please see the modified Figure 2.
- Page 12, Line 426, “Silk sercin has been extensively used in suturing…” This looks like a mistake with silk fibroin. Please check it.
Authors response:
The authors thank the reviewer for noticing this misinformation. The authors revised the literature and the silk fiber (instead of sericin only) is the material that has been used as biomedical suture material for centuries. Modifications were performed (please see line 506 at page 15).
- Page 13, 5.4 should be 5.3.
Authors response:
The authors thank the reviewer for noticing the typo. The numeration of the sub-sections was rearranged.
Round 2
Reviewer 2 Report
Polymers #1973912 – for authors
Reviewer 2 - Second report
#1. The authors responded that the submission of their manuscript for the special issue dedicated to nasal delivery was not their intention. Indeed, their topic does not have anything to do with the topic of this issue. Until somebody solves this mess, I maintain my statement that there is no justification for this submission to be accepted and included in the special issue.
#2, #3. The authors added some references suggested by me, and fixed some of the problems with referencing, making the introduction a bit more palatable. Also, they added more text in section 5, which improved to a certain extent the contributory value of the manuscript to literature.
Upon further examination, I have ended with more observations as delineated below, which clearly require a re-numbering of the references both in the text and in the list ath end.
#4. At lines 330-332, ref. [77] is about sericin produced by Antheraea mylitta, not by Bombyx mori. As in the abstract only the latter is mentioned, the authors should make clear whether their review include non-Bombyx sericins.
#5. Ref. [12] is identical with ref. [33]. Please fix the problem and re-number the list.
#6. What is ref. [2]? If it is a book in Chinese language, its relevance for the general readership is almost nil. If you want to maintain it, please at least provide more information on the publisher, city etc.
#7. In the references that are representing books (refs [6], [62], [90]) the location of publishers (city) is missing. Please complete.
#8. Refs [75] and [76] are irrelevant because (a) apparently they are PhD theses, therefore practically impossible to get; (b) they are in Portuguese language, which is not exactly a widespread language (spoken by only about 3% of world’s population). The authors shall remove them from the list and use them parenthetically within the text.